# Phototherapy of Alzheimer’s Disease: Photostimulation of Brain Lymphatics during Sleep: A Systematic Review

**DOI:** 10.3390/ijms241310946

**Published:** 2023-06-30

**Authors:** Oxana Semyachkina-Glushkovskaya, Thomas Penzel, Mikhail Poluektov, Ivan Fedosov, Maria Tzoy, Andrey Terskov, Inna Blokhina, Viktor Sidorov, Jürgen Kurths

**Affiliations:** 1Department of Physics, Humboldt University, Newtonstrasse 15, 12489 Berlin, Germany; juergen.kurths@pik-potsdam.de; 2Department of Biology, Saratov State University, Astrakhanskaya 82, 410012 Saratov, Russia; thomas.penzel@charite.de (T.P.); fedosov_optics@mail.ru (I.F.); dethaos@bk.ru (M.T.); terskow.andrey@gmail.com (A.T.); inna-474@yandex.ru (I.B.); 3Interdisziplinäres Schlafmedizinisches Zentrum, Charité—Universitätsmedizin Berlin, Charitéplatz 1, 10117 Berlin, Germany; 4Department of Nervous Diseases, Sechenov First Moscow State Medical University, Bolshaya Pirogovskaya 2, Building 4, 119435 Moscow, Russia; polouekt@mail.ru; 5Company “Lazma” for Research and Production Enterprise of Laser Medical Equipment, Kuusinena Str. 11, 123308 Moscow, Russia; victor.v.sidorov@mail.ru; 6Department of Complexity Science, Potsdam Institute for Climate Impact Research, Telegrafenberg A31, 14473 Potsdam, Germany

**Keywords:** Alzheimer’s disease, phototherapy, sleep, photostimulation, mechanisms, brain lymphatic system, brain drainage, beta amyloid, smart sleep technologies, neurodegenerative diseases

## Abstract

The global number of people with Alzheimer’s disease (AD) doubles every 5 years. It has been established that unless an effective treatment for AD is found, the incidence of AD will triple by 2060. However, pharmacological therapies for AD have failed to show effectiveness and safety. Therefore, the search for alternative methods for treating AD is an urgent problem in medicine. The lymphatic drainage and removal system of the brain (LDRSB) plays an important role in resistance to the progression of AD. The development of methods for augmentation of the LDRSB functions may contribute to progress in AD therapy. Photobiomodulation (PBM) is considered to be a non-pharmacological and safe approach for AD therapy. Here, we highlight the most recent and relevant studies of PBM for AD. We focus on emerging evidence that indicates the potential benefits of PBM during sleep for modulation of natural activation of the LDRSB at nighttime, providing effective removal of metabolites, including amyloid-β, from the brain, leading to reduced progression of AD. Our review creates a new niche in the therapy of brain diseases during sleep and sheds light on the development of smart sleep technologies for neurodegenerative diseases.

## 1. Introduction

Alzheimer’s disease (AD) is a brain pathology that is accompanied by progressive memory loss. From 1990 to 2019, the global number of people with AD increased by 117% [1,2,3]. The number of people over 65 years old with AD doubles every 5 years [4]. It is estimated that if an effective therapy of AD is not found, the incidence of AD will triple by 2060 [4]. The large number of clinical studies have failed to show effective pharmacological treatment of AD [5,6,7,8]. Indeed, ENGAGE phase 3 randomized clinical trials of aducanumab obtained in 3285 patients with AD report different consequences associated with therapy, including edema (35.2%), headache (46.6%), confusion (14.6%), dizziness (10.7%), nausea (7.8%), microhemorrhage (19.1%), and superficial siderosis (14.7%) [8].

Photobiomodulation (PBM) is a non-pharmacological therapeutic approach based on the use of red or near-infrared light that has shown very promising results in the treatment of AD in pilot clinical and animal studies [9,10,11,12,13,14,15,16,17,18,19,20]. PBM has been recognized as safe by the U.S. Food and Drug Administration (FDA) because PBM is a non-invasive method without any major side effects. Many publications suggest that the mechanisms of PBM underlie the increase in metabolism and microcirculation of brain tissue, neuroprotection, and reduction in oxidative stress and inflammation [12,17,18,20]. However, it was recently discovered that PBM effectively stimulates lymphatic removal of wastes and toxins, including amyloid-β (Aβ), from the brain [21,22,23,24,25,26,27,28]. Furthermore, PBM affects lymphatic delivery of drugs and nanocarriers to the brain [29,30,31,32].

Recently, the phenomenon of activation of the lymphatic drainage and removal system of the brain (LDRSB) during deep sleep was discovered [21,33,34]. The relationship between the LDRSB and sleep was first described by Xie et al. [33]. This research group compared the brain influx of tracers in mice during sleep, awake, and under anesthesia. The authors clearly demonstrated a 95% reduction in Aβ removal from the brain during wakefulness and the activation of this process during deep sleep due to a 60% increase in the interstitial fluid (ISF) space. Later, Fultz et al. reported that deep sleep is accompanied by high oscillation of cerebral spinal fluid (CSF) in the human brain [34]. There is a hypothesis that the LDRSB is the main driving factor for the removal of metabolic wastes from the sleeping brain [35]. According to this hypothesis, during deep sleep, the size of perivascular spaces and the volume of ISF increase, promoting the removal of metabolites from brain tissue. During wakefulness, the size of the perivascular spaces decreases and the exchange between fluids and brain tissue is suppressed. The similar changes are observed during the development of AD, which is associated with suppression of the LDRSB functions, leading to accumulation of Aβ in the brain tissue.

A few years ago, the idea of using PBM during sleep was proposed as a promising new direction in the treatment of AD [21,35,36,37,38]. Encouraged by this idea, in this review, we discuss how PBM of LDRSB during sleep could be used as a new innovative treatment tool in the therapy of AD. The first session provides information about a strong correlation between the LDRSB and AD pathology. Emerging evidence suggests that the LDRSB is a promising therapeutic target for AD and the development of methods for stimulation of the LDRSB functions could ameliorate cognitive dysfunction in AD and may contribute to progress in the therapy of AD. The second session is devoted to the benefits of PBM for AD as a non-invasive, non-pharmacological, and safe therapeutic approach. Data from animal and clinical studies show that PBM accelerates clearance of Aβ and tau from the brain and could ameliorate cognitive dysfunction in subjects with AD. The third session is focused on PBM during sleep as a new trend in the therapy of AD. The LDRSB and removal of Aβ from the brain is activated at night. Data from animal studies clearly show that nighttime application of PBM can result in a better therapeutic effect in AD mice than PBM during the day. At present, no commercial device for monitoring PBM during sleep is available. We present a concept of such a device and strategy for the development of sleep-PBM technologies to augment the LDRSB and clearance of toxins from the sleeping brain. We suggest that PBM during sleep can be a novel approach to therapy of AD and also for treatment of other brain pathologies associated with impaired LDRSB, including Parkinson’s disease, intracranial hemorrhages, brain trauma, and glioblastoma.

## 2. Methods

### 2.1. Literature Search

The research topic of this review was PBM of AD with a focus on PBM-mediated augmentation of the LDRSB functions during sleep as a strategy for targeting AD.

For this research topic, to capture as many relevant citations as possible, the keywords referring to PBM of interest (photobiomodulation and Alzheimer’s diseases) were combined with the following keywords referring to the LDRSB: meningeal lymphatics, sleep, lymphatic drainage, and clearance. Accordingly, the search engine of the U.S. National Institutes of Health (PubMed) and on the clinicaltrials.gov website (ClinicalTrials.gov is a database of privately and publicly funded clinical studies conducted around the world) was used to retrieve the relevant studies with the following search terms: “photobiomodulation + Alzheimer’s diseases”; “photobiomodulation + the meningeal lymphatics”; “photobiomodulation + sleep”; “photobiomodulation + lymphatic drainage and clearance”; “Alzheimer’s diseases and sleep”; “Alzheimer’s diseases + the meningeal lymphatics”; “Alzheimer’s diseases + lymphatic drainage and clearance”; “the meningeal lymphatics + sleep”; “lymphatic drainage and clearance + sleep”. These search terms retrieved a total of 1471 hits (82 + 4 + 40 + 3 + 1250 + 30 + 24 + 25 + 13 studies). Given the limited information about PBM for healthy volunteers and patients with AD, this review includes studies from the period 2008–2023 and focuses on those investigations into the clinical benefit of using PBM to treat AD.

### 2.2. Inclusion and Exclusion Criteria

Only experimental and clinical trials as well as case reports assessing the effects and mechanisms of PBM for AD were considered. Meta-analyses, conference abstracts, duplicates, or non-English papers were excluded. Furthermore, papers discussing sleep disturbances during the development of AD, PBM used for other brain diseases or for improvement of sleep quality, and other types of therapies for AD were excluded. The total number of papers for further screening was thereby reduced to 133, addressing the LDRSB as a therapeutic target for AD, PBM for AD in basic and clinical studies, and PBM of LDRSB during sleep as a new trend in the therapy of AD.

## 3. Results

We collected 1471 articles from the PubMed database and the clinicaltrials.gov website, including reference lists from previous relevant reviews. After removing duplicates, 973 items remained. We identified 301 records via title screening, and the abstracts of these records were screened more comprehensively. A total of 173 articles were selected for a full-text review, and 133 records were included in the review. The details of excluded trials are shown in Figure 1.

### 3.1. LDRSB as a Therapeutic Target for AD

A growing body of evidence suggests that the meningeal lymphatic vessels (MLVs) as a part of the LDRSB are pathways for the clearance of cell debris, toxic molecules, and wastes from the central nervous system (CNS) [39,40,41,42,43,44,45,46]. It has recently been discovered as an important role of MLVs in the removal of amyloid-β (Aβ) from the brain tissue [38,39]. The accumulation in the brain of Aβ species and plaques is considered to be a biomarker of AD, which precedes neuronal dysfunction and clinical manifestations of AD by up to 20–30 years [47,48,49]. In 1988, Joachim et al. identified Aβ depositions in the meninges of patients with AD [50]. Later, in 2018, Da Mesquita et al. clearly demonstrated that MLVs are tunnels for the clearance of Aβ from the brain [39]. This group showed that photo-injury of MLVs in transgenic mice with AD reduces clearance of Aβ from the mouse brain. The mechanisms underlying Aβ deposition in the brain and the meninges of subjects with AD are still not fully understood. The age-related decrease in recirculation of brain fluids, including the cerebral spinal fluid (CSF) and the interstitial fluid (ISF), is thought to be partly responsible for the accumulation of Aβ in the brain tissue [51,52]. The decline in functions of MLVs with age is associated with reduced excretion of Aβ from brain parenchyma that also contributes to neuronal loss and behavioral changes [52,53,54,55,56]. Therefore, augmentation of functions of MLVs is considered to be a promising therapeutic target for preventing or delaying AD [39,40,57].

The importance of MLVs’ function in the treatment of AD has been demonstrated with immunotherapy [58,59]. This method involves the use of anti-Aβ monoclonal antibodies to reduce Aβ accumulation and to increase Aβ clearance via activation of microglial phagocytosis of this protein. The effects of Aβ immunotherapy are variable in different clinical studies [5,6]. The anti-Aβ monoclonal drug aducanumab was recently approved by the FDA for therapy of patients with AD. However, the approval has sparked a contentious debate over whether the drug is effective [7]. Furthermore, the EMERGE and ENGAGE phase 3 randomized clinical trials of aducanumab obtained in 3285 participants report different symptoms associated with therapy, including edema (35.2%), headache (46.6%), confusion (14.6%), dizziness (10.7%), nausea (7.8%), and microhemorrhage and superficial siderosis in 197 participants (19.1%) and 151 participants (14.7%), respectively [8]. Thus, the use of aducanumab has not been reported as safe.

The latest research has provided a new insight into Aβ immunotherapy of AD via our better understanding of MLVs in regulation of the brain’s immune system [58]. Da Mesquita et al. studied how photo-injury of MLVs and Aβ immunotherapy (m-aducanumab and mAb158, anti-Aβ protofibril monoclonal antibody) can influence AD progression in 5xFAD mice. They demonstrated that Aβ immunotherapy was not effective in reducing Aβ deposits in the brains of mice with the injured MLVs compared with the intact MLVs [58]. Conversely, stimulation of MLVs by intracisternal injection of vascular endothelial growth factor C (VEFG-C), causing lymphangiogenesis, provides for Aβ clearance and improves Aβ immunotherapy. Wen et al., using the APP/PS1 mouse AD model, also show that VEFG-C injection reduces the accumulation of Aβ and alleviates the cognitive deficit [60]. The number of CD8+ “killer” (cytotoxic) and CD4+ “helper” T cells in the meninges significantly increases in old 5xFAD mice with a decrease in MLV coverage that can be improved by VEGF-C via restoring the MLVs’ function [58,59].

These data highlight the potential of VEGF-mediated restoration of the MLVs’ function of improving neuroinflammation and cognitive status in AD subjects. Thus, the effectiveness of Aβ immunotherapy depends on the functional status of MLVs, which is significantly reduced in AD subjects and with age. This can partly explain differences in the reported efficacy of Aβ immunotherapy in humans. These results also provide a new opportunity to further study the effectiveness of Aβ immunotherapy combined with VEGF-C for AD therapy using the different AD models.

In addition, DSCR1, Down syndrome critical region gene 1, regulates the MLV functions in AD [61]. The over-expression of DCSR1 is associated with an increase in clearance of Aβ via MLVs that contribute to an improvement of cognitive function in AD [29]. Wu et al. reveal that borneol, a bicyclic monoterpene belonging to the class of camphene, which is sourced from Blumea balsamifera, improves lymphatic clearance of Aβ as well as other macromolecular polymers in the weight range of 2–45 KD [62].

In summary, current evidence shows a strong relationship between LDRSB and AD pathology. These observations suggest that targeting LDRSB may represent a promising therapeutic strategy for AD. The enhancement of MLV drainage by VEGF-C, DSCR1, and Aβ immunotherapy to accelerate Aβ clearance from the brain could ameliorate cognitive dysfunction in AD [58,60,61]. However, these lymphatic drainage improvement strategies are too far off being introduced in routine clinical practice due to the difficulties of intra-cisterna magna delivery of VEGF-C or high off-targets and low controllability. Therefore, new ideas for stimulation of transport capacity of LDRSB, including clearance of Aβ and tau protein from the brain, are strongly required.

### 3.2. PBM for AD in Basic and Clinical Studies

#### 3.2.1. PBM for AD In Vitro Studies

There are a number of studies showing PDT-mediated reduction in Aβ(42) aggregation in different cell lines. Sommer et al. demonstrated that irradiation with moderately intense 670 nm laser light reduces Aβ aggregates in human neuroblastoma cells [63]. Yang et al. reported that a 632.8 nm laser was capable of suppressing cellular pathways of Aβ-induced oxidative stress and inflammatory responses in primary rat astrocytes [64]. Liang et al. found that a 623.8 nm laser attenuates Aβ-induced cell apoptosis through the Akt/GSK3β/β-catenin pathway [65]. They showed that PBM activates Akt, which interacts with GSK3β and phosphorylates it on Ser9 in the presence of Aβ(25–35), leading to the inhibition of GSK3β and promoting cell survival upon treatment with Aβ(25–35). Zhang et al. discovered reduction in Aβ(25–35)-induced cell apoptosis by a 632.8 nm laser through promoting Akt-dependent Yes-associated protein cytoplasmic translocation [66]. The downregulation of brain-derived neurotrophic factor (BDNF) in the hippocampus is a consequence of the progression of AD and Aβ-induced neurotoxicity. There is a hypothesis that BDNF plays an important role in neuronal survival and contributes to rescuing dendrite atrophy and cell loss in AD. Meng et al. clearly showed that a 632.8 nm laser protects Aβ-treated hippocampal neurons and cultured APP/PS1 mouse hippocampal neurons via upregulation of BDNF in both [67]. Zinchenko et al. investigated 1267 nm laser-induced extravasation of Aβ (1–42) through the model of the blood–brain barrier as a mechanism of PBM for AD [25].

#### 3.2.2. PBM for AD in Animal Studies

In 2011, De Taboada et al. presented promising results demonstrating the effective transcranial PBM therapy of Aβ-precursor transgenic mice by using a 810 nm laser (3 sessions per week for 6 months) [68]. They found that Aβ plaques and inflammatory markers were reduced significantly in the brain tissue in a laser dose-related manner (4.8 J/cm^2^−48 J/cm^2^ on the skull and 1.2 J/cm^2^−12 J/cm^2^ on the cortex). These therapeutic effects attenuated the progression of AD, leading to an improvement of cognitive functions in mice. The dose-dependent reduction in Aβ plaques in the brain of TASTPM transgenic mice was also shown by Grillo [69]. Purushothuman et al. demonstrated the beneficial effect of transcranial PBM (670 nm LEDs, 4 J/cm^2^, 20 min per session for 4 weeks) in two mouse models of AD, such as the K369I tau transgenic model and the APPswe/PSEN1dE9 transgenic model [70,71]. In the tau model, PBM restored the expression of mitochondrial cytochrome c oxidase in neurons as well as decreased markers of neurofibrillary tangles and oxidative stress, including 4-hydroxynonenal and 8-hydroxy-2-deoxyguanosine, to the normal values in the neocortex and the hippocampus. In both tau and Aβ models, PBM reduced the level of tau protein and Aβ plaques in the brain tissue. Using the 5XFAD model of AD, Farfara et al. treated mice by PBM (810 nm, 1 J/cm^2^ applied to the tibia via a small incision of the skin) for 4 weeks with 10-day intervals starting at the age of 4 months. The results revealed a 68% decrease in Aβ plaques in the brain, which was associated with better performance in the object recognition and fear conditioning tests in the PBM group compared with the sham group [72]. Oron et al. discussed these results in their review as the new strategy of the PBM therapy of AD via the secondary PBM effects through photostimulation of proliferation of mesenchymal stem cells [73]. Lu et al. showed the therapeutic effects of PBM (810 nm, 2 min per session for 5 days) in rats with the injected model of AD [74]. They showed that PBM inhibited neurodegeneration in the hippocampus, leading to reduction in Aβ accumulation in the brain and improvement of recognition memory. They also found positive PBM effects on the mitochondrial functions, including enhancement of mitochondrial antioxidant expression, the level of cytochrome c oxidase activity and ATP, and suppression of apoptosis, reactive gliosis, and inflammation. De Luz Eltchechem et al. also demonstrated a significant reduction in Aβ plaques in the hippocampus and an improvement of cognitive, behavioral, and motor skills in rats with the injected model of AD after PBM therapy (627 nm, 7 J/cm^2^, 1.5 min for session for 3 weeks) [75]. Brivelet et al. reported similar therapeutic effects of PBM in Swiss mice with the injected model of AD after PBM (LED, 625 nm, 10 Hz, 8.4 J/cm^2^, 10 min for 7 days) [76]. Cho et al. used the transcranial PBM treatment (610 nm, 1.7 mW/cm^2^, 2.0 J/cm^2^, 20 min per session three times per week for 14 weeks) of 5XFAD mice and found that PBM in early stages of AD reduced Aβ accumulation in the brain and neuronal loss and alleviated cognitive dysfunction [77]. Zhang et al. presented the results of transcranial PBM (632.8 nm, 2 J/cm^2^, 10 min per session for 30 days) in double transgenic mice (APPswe/PSENdE9) and showed that PBM reduced Aβ levels in the brain and improved spatial learning and memory deficits [78]. Zhinchenko et al. reported the optimal 1267 nm laser dose (32 J/cm^2^) for reduction in Aβ accumulation in the brain and restoration of recognition memory in mice with the injected model of AD after 10 days of PBM therapy [27].

#### 3.2.3. PBM for AD in Clinical Studies

Several studies demonstrate that PBM improves memory, attention, sleep quality, and emotional state in healthy humans. Table 1 presents data on the parameters of PBM used in healthy volunteers involved in studies investigating the PBM effects on mental functions and memory. There is a hypothesis that the mechanisms of PBM effects on a healthy brain can be an increase in cerebral perfusion, brain energy, and oxygen metabolism, which are necessary to maintain normal cognitive function [12,17,20,79]. Barret and Gonzalez-Lima used a 1064 nm laser in a placebo-controlled study and reported the beneficial PBM effects on executive functions, attention, and memory in healthy volunteers 2 weeks after PBM [80]. Blanco et al. demonstrated a superior Wisconsin Card Sorting Test performance after PBM (1064 nm) compared with the control group [81]. Chan et al. observed an improvement of reaction time evaluated by the Eriksen flanker test and mental flexibility assessed by the category fluency test in older (over 60 years) healthy volunteers after a single session of PBM (a mix of 633 nm and 870 nm) [82]. Wu et al. investigated the PBM effects (830 nm) on the pattern of electrical brain activity using electroencephalogram (EEG) in forty normal healthy male subjects [83]. They observed for 10 min after PBM an increase in the power of alpha rhythms and theta activities in the occipital, parietal, and temporal areas (mainly in the posterior head regions). The authors suggested that the PBM effects on the EEG changes are comparable to those in meditation. Jahan at el. suggested that transcranial PBM (850 nm) on the right prefrontal cortex changed the brainwaves (decreases a delta power) and had a beneficial effect on cognitive performance as well as improving attention and alertness in 30 healthy young adult participants [84].

The clinical studies of the PBM effects for AD are still limited [9,10,11,19,85]. Table 2 summarizes the results of recent clinical studies on the therapeutic effects of PBM in patients with AD. Salmarche et al. investigated five patients with mild to moderately severe dementia or possible AD and demonstrated that home transcranial–intranasal PBM (810 nm) therapy for 12 weeks can improve cognition and functional abilities for daily living [86]. Chao et al. also reported that home transcranial PBM (801 nm) for 12 weeks significantly improved cognitive and behavioral functions and increased cerebral microcirculation in four patients with mild-to-moderate dementia [87]. Later, the same group demonstrated a decrease in the CSF levels of Aβ42, tau, and neurofilament light chain in seven patients with AD 16 weeks after home transcranial–intranasal PBM (810 nm) therapy [88]. The reduction in the CSF levels of Aβ42 was also observed after therapy of patients with AD, using flickering lights at gamma frequency for 8 weeks [89]. Zomorrodi et al. presented similar results using home PBM (810 nm) treatment of one patient with moderate AD [90]. Salehpour at el. reported positive effects of a transcranial–intranasal PBM treatment (a mix of 635 nm and 810 nm) of a single case on cognitive enhancement and reversal of olfactory dysfunction [91]. Maksimovich used low-energy lasers in the visible region of the spectrum for trans-catheter treatment of AD and showed that PBM reduces the effects of dyscirculatory angiopathy of AD and improves cerebral microcirculation and metabolism, leading to dementia regression and cognitive impairment reduction [15]. However, in another study on 11 patients with dementia, no significant differences were found between the PBM (1060–1080 nm and the sham groups, probably due to the small number of patients in the PBM group (*n* = 6) and in the sham group (*n* = 5) [92]. However, the authors reported a trend of improvement in executive functioning, praxis memory, visual attention, and task switching in participants receiving PBM for 4 weeks. Later, the same group repeated the clinical studies of the PBM therapy for AD on a larger group of 47 patients and showed that PBM has positive cognitive, executive, and mood changes in participants with dementia that improve their quality of life [93]. Further analysis of these human data revealed no gender difference in the effectiveness of the PBM therapy of dementia [93]. Horner et al. analyzed the effectiveness of the transcranial PBM (810 nm) therapy of one patient with AD and type 2 diabetes and showed an improvement of cognitive function and restoration of mitochondrial function after 10 weeks of treatment [94].

#### 3.2.4. PBM for Other Brain Diseases

Parkinson’s disease (PD) is a progressive brain disease that leads to degeneration of neurons and affects the control of body movements. PBM for PD has received much attention as one of the most promising methods to treat this neurodegeneration in the brain [10,18,79,95]. Several clinical in vitro (the blood samples of 10 PD patients) and in vivo (*n* = 70 PD patients) studies reported the therapeutic effects of PBM (632.8 nm; 500 mW/cm^2^) on enzyme activity of monoamine oxidase B (MAO B), Cu/Zn-superoxide dismutase (Cu/Zn-SOD), Mn-SOD, and catalase in blood from PD patients [96,97]. PBM consisted of five daily 20 min sessions of blood irradiation (intravenous irradiation in vivo), and the blood was analyzed on day 3 after the last irradiation session. PBM significantly improved neurological status of PD patients assessed by Unified Parkinson’s Disease Rating Scale. The improvement was accompanied by normalization of MAO B and Cu/Zn-SOD activities. The effects of PBM on blood enzymes were much stronger after in vivo application of PBM than after in vitro blood irradiation.

Reduced axonal transport is one of the mechanisms underlying the progressive loss of dopaminergic nerve terminals in PD. The ATP synthesis in mitochondria supports axonal transport and contributes to the survival of neuronal cells. In PD subjects, mitochondria in brain tissue are metabolically and functionally suppressed. Trimmer et al. demonstrated that PBM (810 nm, 50 mW/cm^2^ for 40 s) improves axonal transport in model human dopaminergic neuronal cells [98]. They measured the velocity of movement of labeled mitochondria in human transmitochondrial cybrid neuronal cells bearing mitochondrial DNA from patients with PD and healthy volunteers. They found that the velocity of mitochondrial movement in neuronal cells was significantly reduced in the PD group vs. the control group. However, 2 h after PBM, the velocity of mitochondrial movement in PD returned to normal values, while it was unaltered by PBM in the control groups. The authors suggest that PBM can be seen as a novel strategy in the therapy of PD.

Amyotrophic lateral sclerosis (ALS) is another neurodegenerative disorder that leads to gradual muscle paralysis, resulting in respiratory failure and death within 1–5 years after onset of clinical signs. The mitochondrial dysfunction is involved in the pathogenesis of ALS. Moreover, oxidative stress is also thought to play an important role in ALS.

Moges et al. examined the effects of PBM (810 nm, 140 mW, 1.4 cm^2^ for 120 s, for three consecutive days every week for 51 days) on the survival of motor neurons in the G93A SOD1 transgenic mice (ALS model) [99]. The PBM was applied in three places: the primary motor cortex, the cervical enlargement of the spinal cord, and the lumbar enlargement of the spinal cord. Despite the fact that motor function in G93A SOD1 mice was improved after PBM, however, PBM was ineffective in altering progression of ALS. Nevertheless, these findings have potential implications for the conceptual use of PBM to treat other neurodegenerative diseases associated with mitochondrial dysfunction.

A PBM has been shown to be effective in therapy for other brain diseases as well. Muili et al. found that PBM (670 nm, 28 mW/cm^2^ for 3 min, once daily irradiation) ameliorates symptoms in a mouse model of multiple sclerosis [100]. Leisman et al. reported improvement of autism spectrum disorder in children and adolescents of 5–17 years of age after PBM treatment (635 nm, 15 mW for 5 min, a 4-week course of PBM) [101]. PBM can also prevent hypoxic–ischemic brain injuries by maintaining mitochondrial function, decreasing oxidative stress and inhibiting neuronal apoptosis. Indeed, Tucker et al. found that PBM (808 nm, 25 mW/cm^2^ for 2 min, 7 day treatment) prevents ischemic injury to neurons after cerebral ischemia caused by cardiac arrest [102]. Yang et al. using a model of neonatal hypoxic–ischemic encephalopathy in rats show that PBM (808 nm, 100 mW/cm^2^ for 2 min) improves ischemic-induced brain injury [103]. Thus, the results discussed above clearly demonstrate that PBM can significantly attenuate cognitive impairment, neuron loss, and dendritic and synaptic injury, and restore ischemic-induced mitochondrial dynamics in various brain diseases.

### 3.3. PBM of the LDRSB during Sleep as a New Trend in the Therapy of AD

Sleep is considered to be a novel biomarker and promising therapeutic target for AD and dementia [35,36]. Emerging evidence suggests that sleep disorder is an independent risk factor of AD and the analysis of sleep quality can be an informative approach to screen for AD [104,105,106,107,108,109,110,111,112]. It is well known that AD has two interrelated characteristics, such as poor sleep and increased Aβ deposition in the brain [107,109]. In a recent long-term study on a large group of 8000 volunteers aged 50–60 years, it has been discovered that chronic sleep deficit (less than 6 h) for 25 years can lead to the development of dementia [113]. However, even one night without sleep is associated with a rise in Aβ level in the brain of young and healthy people [114,115].

There is a growing body of evidence suggesting that sleep is accompanied by activation of the LDRSB, leading to Aβ clearance from the brain [21,33,34,35,36,116,117,118]. Indeed, the CSF level of Aβ42 is increased at night before sleep and is decreased in the early morning after sleep in both humans and animals [114]. However, not all sleep is equivalent to activation of the LDRSB. Recent studies clearly demonstrate that only slow-wave activity (SWA, 0–4 Hz) during non-rapid eye movement (NREM) sleep is optimal for the LDRSB [119]. In 2013, Xie et al. reported that the LDRSB function is highest during NREM sleep and lowest during wakefulness [33]. This group found a 95% reduction in Aβ removal from the brain in awake mice but a significant activation of Aβ clearance during NREM sleep associated with a 60% increase in the interstitial space [33]. Later, Benveniste et al. confirmed that sleep is accompanied by the expansion of the interstitial space, probably due to an increase in the flow of ISF for optimization of removal of metabolites [120]. Figure 2a illustrates our experimental results, demonstrating an activation of drainage of the mouse brain along the perivascular space during deep sleep. The SWA disturbances reflect the AD pathology in both humans and animals [109,121,122,123,124]. In humans, impaired memory consolidation associated with AD progression is strongly correlated with decreased SWA time [125,126,127]. The mechanisms underlying the Aβ-mediated reduction in the SWA dynamic and a decrease in the LDRSB functions remain unknown. The Aβ accumulation in the brain leads to hyperactivity of the cortical neurons and synaptic abnormalities that can be a possible mechanism underlying the SWA deficit in AD subjects [128,129,130,131,132,133]. The insufficient sleep is related to an increase in the interstitial noradrenaline level, leading to a reduction in the astrocytic volume and the cerebral vasoconstriction that can be another mechanism of a suppression of the LDRSB function and removal of wastes from the brain [33,134,135]. The horizontal body posture during sleep plays an important role in the LDRSB function, contributing to a better Aβ clearance from the brain [136].

## 4. Discussion

The global number of elderly people with AD is progressively increasing worldwide. However, pharmacological therapies for AD have failed to show effectiveness and safety [5,6,7,8]. Therefore, the development of alternative non-pharmacological therapeutic methods for AD is highly relevant for medicine. This review is focused on PBM as a non-invasive, non-pharmacological, safe, and promising approach for treating AD and other brain pathologies. Since the treatment of AD is incurable, therapy aimed at reducing the progression of the disease should be applied throughout the life of patients. In this respect, PBM technologies are the most promising, since they are already widely used in the clinical field for the treatment of various brain diseases [137,138,139,140].

The rapidly increasing number of studies demonstrates the importance of the LDRSB in resistance to AD pathology and suggests that the development of methods for augmentation of the LDRSB functions may contribute to strong progress in the therapy of AD [21,27,35,36,39]. Traditionally, it was thought that the mechanisms of therapeutic effects of PBM underlie the increase in metabolism and microcirculation of brain tissue, neuroprotection, and reduction in oxidative stress and inflammation [12,17,18,20]. However, it was recently discovered that PBM also can augment the LDRSB functions, providing effective removal of metabolites, including Aβ, from the brain, leading to reduced AD progression [21,22,23,24,25,26,27,28,29,30,31,32,33,34,35].

Since the LDRSB and removal of Aβ from the brain are activated at night, it is logical to expect that PBM for AD during sleep will be more effective than PBM during day. Indeed, the pilot animal data clearly demonstrate that the night-course of PBM has better therapeutic effects in AD mice compared with the day-course of PBM [21] (Figure 2b–d). These pioneering results open up new perspectives for PBM therapy of AD during sleep, which is of growing interest to researchers [35,36,37,38].

Typically, in experimental and clinical studies, PBM is performed in awake subjects [9,10,11,12,13,14,15,16,17,18,19,20,68,69,70,71,72,73,74,75,76,77,80,81,82,83,84,85,86,87,88,89,90,91,92,93,94]. In our recent review, we suggested that there are no commercial devices for simultaneous PBM and sleep monitoring, which significantly limits the clinical investigations [36]. The development of sleep-PBM technologies for stimulation of the LDRSB during deep sleep is urgently needed and could become a breakthrough step in the progress of AD therapy in humans (Figure 3).

The mechanism by which PBM affects the lymphatics has not been sufficiently explored. Data from animals demonstrate that PBM stimulates the drainage and cleaning functions of the lymphatics via PBM-mediated activation of lymphatic contractility [29]. Li et al. also report that PBM increases lymphangion contraction [45]. Nitric oxide (NO) plays an important role in regulation of relaxation of the lymphatic vessels after their contraction [141,142]. The contraction cycle of the lymphatic vessels is a combination of mechanical and conducted electrophysiological events. The relaxation phase is a component of locally generated NO in the lymphatic endothelial cells (LECs) in response to transiently elevated shear forces. The NO is a vasodilator that acts via stimulation of soluble guanylate cyclase to form cyclic-GMP, which activates protein kinase G, causing the opening of calcium-activated potassium channels and reuptake of Ca^2+^. The decrease in the concentration of Ca^2+^ prevents myosin light-chain kinase from phosphorylating the myosin molecule, leading to relaxation of the lymphatic vessels after their contraction. Based on these facts, Li et al. studied the NO production in in vitro experiments on the mesenteric LECs without and after PBM. These results revealed a significant increase in the 24 h accumulation of NO in the cell culture medium after PBM compared with the accumulation of NO produced by LECs without PBM [45]. These results suggest that PBM stimulates the LDRSB functions by influencing the mechanisms of regulation of lymphatic pumping via regulation of the contraction cycle and phase of relaxation of the lymphatic vessels by activation of the intracellular NO production. NO-induced relaxation allows for diastolic filling of the lymphatic vessels and thus prepares them for their next contraction [143,144,145].

How PBM can improve cognitive function in subjects with AD remains poorly understood. Emerging evidence indicates the important homeostatic and pathophysiological roles of brain lymphatics in the progression of AD [21,39,40,146,147]. Indeed, genetic and pharmacological disruption of MLVs results in less drainage of CSF and ISF to the cervical lymph nodes [31,39]. Such disruption also results in cognitive impairment and behavioral alterations [39]. Increasing lymphangiogenesis of MLVs via administration of the vascular endothelial growth factor improves the drainage of macromolecules to the cervical lymph nodes of elderly mice [58,148,149,150,151]. Finally, disruption of MLVs worsens mouse models of AD. Collectively, these findings suggest that dysfunction of the LDRSB might provide an important contribution to age-related cognitive decline and neurodegenerative disease [146,147].

We assume that the mechanisms of the PBM therapy of AD during sleep might be different from those during wakefulness, which requires careful and detailed research [35,36,37,38]. Further animal and clinical studies of night PBM of AD will shed light on optimal parameters of PBM, including light wavelength, power density and intensity, treatment duration, and comfort light application on the head in sleeping subjects that are critical for the successful PBM therapy of AD.

Since there are few studies on the use of PBM for healthy volunteers and patients with AD, this limits information about reproducibility of reported protocols, parameters, intensity, duration, and intervals of PBM in humans as well as about an optimal optical window and mechanisms responsible for PBM-mediated therapy of AD. There is a wide range of parameters for the transcranial–intranasal PBM in healthy subjects and AD patients (Table 1 and Table 2). Devices with LEDs both in red light (633–635 nm) and in near-infrared wavelength (800–900 nm and 1064–1080 nm) are most popular in PBM for AD. The benefit of an optical window is that the light wavelengths (600–1200 nm) can effectively penetrate the skin and the skull. Tedford et al. estimated that the wavelength of 808 nm has a maximal penetration depth [152]. Wang and Li proposed that the wavelengths 660 nm and 810 nm are optimal for transcranial PBM [153]. In our recent review, we discussed in detail the depth of penetration of different light wavelengths into the head and brain tissue in humans [154]. The energy density ranges from 10 to 60 J/cm^2^ and pulsing of light is at 10–40 Hz. The duration of PBM is rather small in healthy subjects (typically, single irradiation lasting 2.5–10 min) but is long enough in AD patients (usually 20–25 min per session and for 4–16 weeks but can be from 1 year to 12 years).

A PBM has been recognized as safe by the U.S. Food and Drug Administration (FDA). Red-light therapy is generally considered safe, even though researchers are not exactly sure how and why it works. Maximal permissible exposures of skin surface as well as other conditions for safe use of light radiation are regulated with relevant standards, e.g., ANSI Z136 [155,156]. Even though this type of treatment is generally very safe, some negative effects may occur [157,158,159,160]. Mild visual side effects are not unusual but disappear promptly. Therefore, determining the appropriate dose and timing of light is essential in order to diminish the occurrence of such side effects. However, there are no set rules on how much light to use for the therapy of AD. Nonetheless, the use of PBM for people with drug-resistant non-seasonal depression can result in a hyperactive state and an increase in blood pressure [161]. In these rare cases, light therapy must be reduced or stopped and the condition adequately treated.

We want to emphasize that given the complex pathophysiologic nature of AD, a single therapeutic intervention is unlikely to be a satisfactory response, and that a combination of various interventions is probably critical.

Noninvasive brain stimulation (NIBS) is another strategy for AD treatment [162,163,164,165,166,167,168,169,170,171]. Transcranial magnetic stimulation (TMS) is the most widely used technique for NIBS. TMS consists of delivering short (up to 300 µs) magnetic pulses of high intensity (up to 2.5 Teslas) by a copper-wired coil applied to the scalp. The induced electric field in the brain triggers action potentials, and alters neural activity.

In this respect, TMS has been successfully used in investigating molecular and neurotransmitter dysfunctions, characterizing AD pathology and highlighting biomarkers for the differential diagnosis between AD and other forms of dementia [172,173]. In a transgenic animal model, 14 days of TMS reduces Aβ deposition in the hippocampus and prevents spatial memory loss [161]. Remarkably, such neuroprotective effects are associated with increased CSF-ISF exchange dynamics, as observed by increased CSF glymphatic influx and meningeal lymphatic outflow into the deep cervical lymph nodes, resulting in reduced gliosis and increased neuronal activation in the hippocampus [174]. It is worth noting that a single session of TMS using continuous theta burst stimulation can restore the sleep deprivation-induced reduction in the expression of polarized AQP4 [175] and increase meningeal lymphatic dilation [176], suggesting that TMS increases clearance function by acting on both the glymphatic and meningeal lymphatic pathways. However, the major limitation across TMS lies in the difficulty of comparing its efficacy due to the high variability observed across study protocols [163,165]. Thus, current results on the use of NIBS in AD are encouraging, but there is still a need to better characterize the long-term benefits of stimulation [165].

## 5. Conclusions

In this review, we highlighted the most recent and relevant experimental and clinical studies of the effectiveness of PBM for the therapy of AD and other brain diseases. Increasing evidence for the important role of the LDRSB in resistance to AD has provided new insights into AD progression and laid the foundation for the development of innovative approaches for stimulation of the LDRSB functions. We suggested that PBM might be a promising technology to target the LDRSB. Since the LDRSB and removal of Aβ from the brain are activated at night, we proposed using PBM during sleep as a new trend in AD therapy. We also presented a concept for a device and strategy for the development of sleep-PBM technologies to augment the LDRSB and clearance of toxins from the sleeping brain. Many brain diseases are associated with impaired LDRSB and suppression of removal of toxins from the brain, including PD, intracranial hemorrhages, brain trauma, and glioblastoma [41,42,43,44,45,177]. There are experimental and clinical data suggesting the effective therapy of these brain diseases by PBM [29,31,45,137,138,139,140,178,179]. Therefore, nighttime therapy of brain diseases using PBM and other approaches during deep sleep might herald a new era in neurorehabilitation medicine.

## Figures and Tables

**Figure 1 ijms-24-10946-f001:**
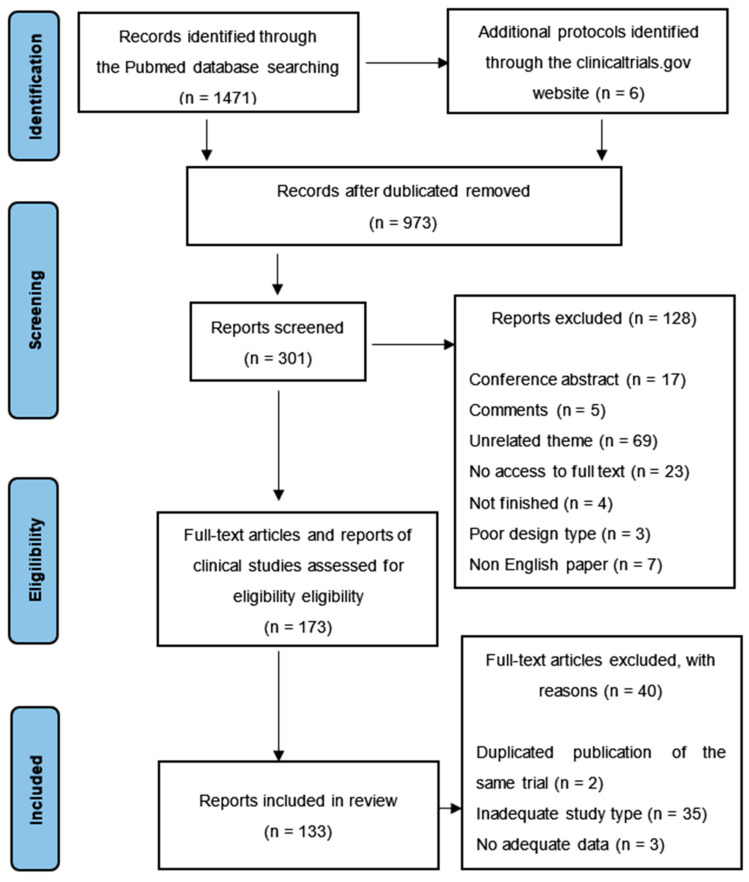
PRISMA flow diagram depicting the study process. n, the number of records; PRISMA, Preferred Reporting Items for Systematic Reviews and Meta-Analyses.

**Figure 2 ijms-24-10946-f002:**
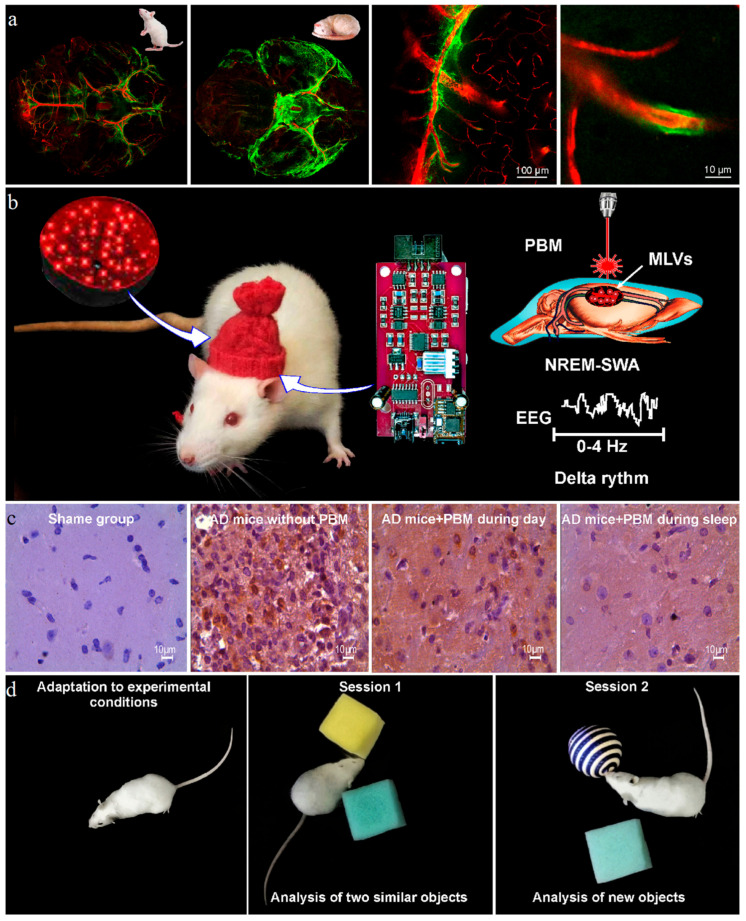
PBM of the LDRSB during deep sleep: (**a**) activation of drainage of the mouse brain during deep sleep along the perivascular space (green, fluorescein isothiocyanate-dextran 70 kDa filled the perivascular spaces); (**b**) schematic illustration of hat-shaped device for simultaneous wireless EEG monitoring and PBM of the LDRSB; (**c**,**d**) animal data demonstrating that PBM of the LDRSB during deep sleep more effectively reduces Aβ accumulation in the mouse brain and restores recognition memory assessed by performance of the novel object recognition test [21].

**Figure 3 ijms-24-10946-f003:**
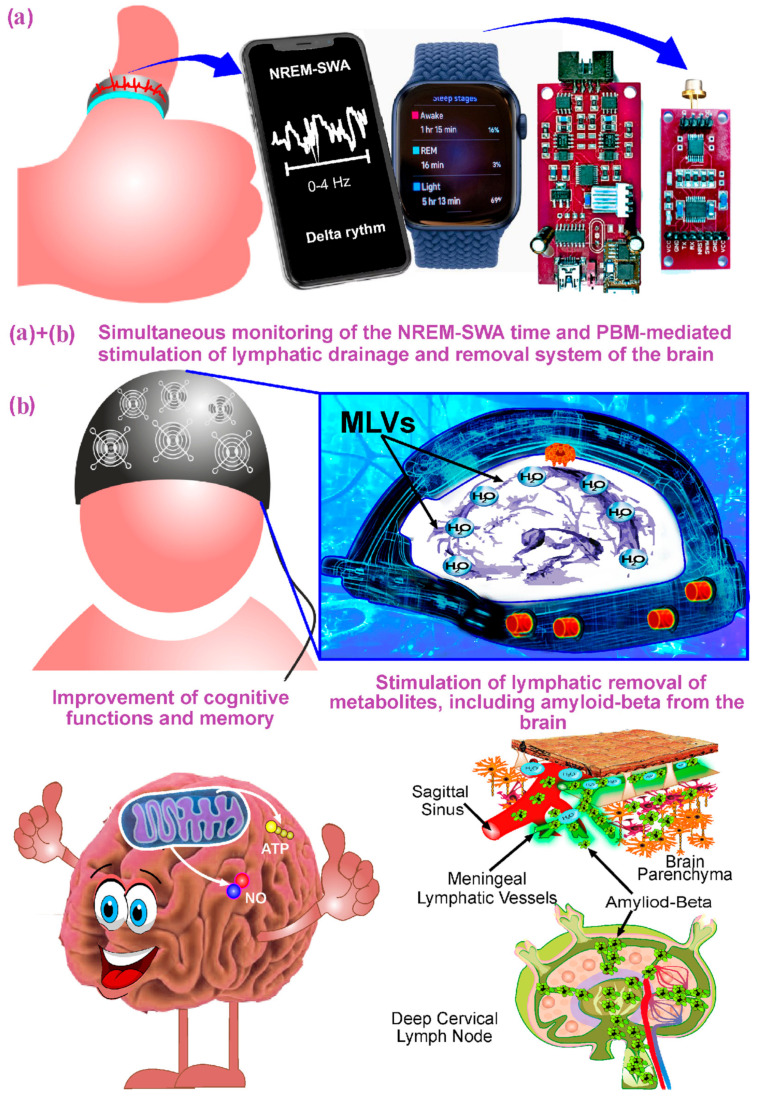
Schematic illustration of promising strategy of the therapy of AD during deep sleep, including simultaneous EEG monitoring of sleep stages (**a**) and PBM of LDRSB (**b**). This innovative technology can be implemented through the creation of software for sending a signal from a wireless sleep-tracking gadget (bracelet, ring, smart watch, etc.) to a PBM device (helmet, cap, bandanas, etc.) [36]. The PBM during deep sleep corresponds to the time of natural night activation of lymphatic clearance of metabolites, wastes, and cellular debris from the brain tissue [21,33,34,100,112,113,114,115] that provides a better therapeutic effect for AD pathology than PBM during the day [21].

**Table 1 ijms-24-10946-t001:** PBM in healthy people.

First Author (Year of Publication) and References	Wavelength (nm) and Irradiation Parameter	Time and Duration of PBM	Groups and Number of Volunteers	Therapeutic Effects
Barret and Gonzalez-Lima (2013) [80]	1064 nm 250 mW/cm^2^, 60 J/cm^2^	Transcranial PBM 8 min, single irradiation	Total 40 healthy volunteers (*n =* 20 in the PBM group *n =* 20 in the sham group)	Improvement of executive functions, attention, and memory 2 weeks after PBM
Blanco et al. (2017) [81]	1064 nm 250 mW/cm^2^, 60 J/cm^2^	Transcranial PBM 8 min, single irradiation	Total 30 healthy volunteers (*n =* 15 in the PBM group; *n =* 15 in the sham group)	Improvement learning assessed by the use of the Wisconsin Card Sorting Test
Chan et al. (2019) [82]	A mix of 633 nm and 870 nm LEDs 44.4 mW/cm^2^, 20 J/cm^2^	Transcranial PBM 7.5 min, single irradiation	Total 30 healthy volunteers (*n =* 15 in the PBM group; *n =* 15 in the sham group)	Improvement of reaction time and mental flexibility during performance of the Eriksen flanker and category fluency tests
Wu et al. (2012) [83]	830 nm 7 mW per diode, 20 J/cm^2^	Transcranial PBM 10 min, single irradiation	Total 40 healthy volunteers (*n =* 20 in the PBM group; *n =* 20 in the sham group)	Increase in alpha rhythms and theta activities in the occipital, parietal, and temporal regions
Jahan et al. (2019) [84]	850 nm 285 mW/cm^2^, 60 J/cm^2^. The total power was 400 mW with a 1.4 cm^2^ irradiation area.	Transcranial PBM 2.5 min, single irradiation	Total 30 healthy volunteers (*n =* 15 in the PBM group; *n =* 15 in the sham group)	Beneficial effect on cognitive performance Improvement in attention and alertness

**Table 2 ijms-24-10946-t002:** PBM for AD in clinical studies.

First Author (Year of Publication) and References	Wavelength (nm) and Irradiation Parameter	Time and Duration of Phototherapy	Groups and Number of Patients	Therapeutic Effects
Saltmarche et al. (2017) [86]	810 nm 14.2 mW/cm^2^ (transcranial), 10.65 J/cm^2^ (intranasal)	Transcranial–intranasal PBM. 25 min every day for 12 weeks.	Five participants with dementia or AD	Improvement of cognition, functional ability in everyday life
Chao (2019) [87]	810 nm 75 mW/cm^2^, 45 J/cm^2^	Transcranial PBM. Once every other day for 20 min for 12 weeks.	Total 8 patients with mild-to-moderate dementia (*n =* 4 in the PBM group; *n* = 4 in the sham group	Improvement cognitive and behavior functions, increase in cerebral perfusion and connectivity between the posterior cingulate cortex and lateral parietal nodes within the default-mode network
Maksimovich (2015 and 2019) [15]	Low-energy lasers in the visible region of the spectrum, 20 mw power	Under local anesthetic, the common femoral artery was catheterized and a thin, flexible fiber-optic (diameter 0.25 to 100 μm) was advanced to the distal sections of the anterior and middle cerebral arteries where PBM was performed, taking 20–40 min in the period from 1 year to 12 years after the first symptoms of AD	Total number 89 with AD (*n* = 46 in the PBM group and *n* = 43 in the sham group)	Improvement in cerebral microcirculation, reduction in dementia and restoration of cognitive functions
Nizamutdinov (2021) [16]	1060–1080 nm, 15,000 mW, 23.1 mW/cm^2^, ~650 cm^2^ per treatment area	Transcranial PBM, two 6 min sessions daily for 8 consecutive weeks	Total 60 patients with dementia *n* = 47 in the PBM group and *n* = 13 in the sham group	Positive cognitive, executive, and mood changes. Improvement in quality of life.
Zomorrodi (2017) [90]	810 nm wavelength, 40 Hz	Transcranial PBM, every 20 min once a night, 6 nights a week for 17 weeks	One patient with moderate AD	Significant improvement in cognition. Outcomes were rapid and significant, noticeable within days, continuous, and sustained over 3 weeks.
Berman (2017) [92]	1100 LEDs set in 15 arrays of 70 LEDs/array with all matched to 1060–1080 nm, 10 Hz with a 50% duty cycle	Transcranial PBM, 6 min daily over 4 weeks	Total 11 patients with dementia (*n* = 6 in the PBM group; *n* = 5 in the sham group)	No significant differences between the PBM group and the control group. A trend of improvement in executive functioning; praxis memory, visual attention, and task switching
Salehpour et al. (2019) [91]	A mix of 635 nm and 810 nm LEDs, 10 Hz	Transcranial–intranasal PBM, 25 min per session twice daily (morning and evening) for 4 weeks	One patient with cognitive impairment and olfactory dysfunction	Cognitive enhancement and reversal of olfactory dysfunction
Horner et al. (2020) [94]	810 nm, 40 Hz; 50% duty cycle 25 mW/cm^2^ (the nasal applicator) 100 mW/cm^2^ (three posterior LEDs) and 75 mW/cm^2^ (the anterior LED)	Transcranial–intranasal PBM, 25 min per session for 10 weeks	One patient with mild AD and type 2 diabetes	Improvement in cognition and restoration of mitochondrial function
The following clinial studies of PBM for AD have been registered on the clinicaltrials.gov website				
Chao (2022) [88]	810 nm at 40 Hz 40–150 mW/cm^2^	Transcranial and intranasal Once every other day for 20 min for 16 weeks	Total 14 patients with mild-to-moderate AD (*n* = 7 in the PBM group; *n* = 7 in the sham group)	Reduction in the CSF levels of Aβ42, tau, and neurofilament light chain
Lah (2020) [89]	Flickering lights at gamma frequency	One hour per day for 8 weeks	Total 10 patients with mild-to-moderate AD (*n* = 5 in the Friker group; *n* = 7 in the sham group)	Improvement in blood flow and reduction in the CSF levels of Aβ42

## Data Availability

Not applicable.

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
