# Peer review of "Phototherapy of Alzheimer’s Disease: Photostimulation of Brain Lymphatics during Sleep: A Systematic Review"

_ijms, 2023, doi:10.3390/ijms241310946_

Round 1

Reviewer 1 Report

This review presents the evidence related to methods for augmentation of the LDRSB functions during sleep as a strategy for targeting AD. The focus is on photobiomodulation as a non-invasive, non-pharmacological, and safe approach. Data from animal studies show that nighttime application of PBM can result in therapeutic effects in AD mice. At present, no commercial device for PBM under sleep monitoring is available. The authors have presented an idea of such a device and strategy for the development of sleep-PBM technologies to augment the LDRSB during deep sleep. This can be a novel strategy for AD and also other brain diseases associated with impaired LDRSB including Parkinson’s disease, intracranial hemorrhages, brain trauma, and glioblastoma. 

The authors are encouraged to add their search strategy for this review to ensure the selection of all relevant literature and assessment of the quality of the studies that are included together with the limitations of this review work. 

Please also add if any safety precautions must be taken into account in the model that has been outlined. 

What will be the expectation of use by patients in terms of intensity, interval, and long-term? repeated, short-term? Please elaborate.

Please proof read the text for the English.

Author Response

Comment: The authors are encouraged to add their search strategy for this review to ensure the selection of all relevant literature and assessment of the quality of the studies that are included together with the limitations of this review work. 

Response: The authors would like to express their sincere gratitude to the referee for the recommendations and great help in improving our paper. We added the description of search strategy for our review (Lines 57-91). All changes in the text of the article are highlighted in yellow.

Comment: Please also add if any safety precautions must be taken into account in the model that has been outlined. 

Response: We improved Discussion and added information about relevant standards of safe use of PBM and its possible side-effects (Lines 391-402).

Authors

Comment: What will be the expectation of use by patients in terms of intensity, interval, and long-term? repeated, short-term? Please elaborate.

Response: We added data on the parameters of PBM used in healthy volunteers and patients with AD (Tables 1 and 2, Lines 374-390, 434-442).

The authors would like to thank the referee again for great help in improving the quality of our article and its possible publication in International Journal of Molecular Science.

Reviewer 2 Report

26 April 2023 

Manuscript ID: ijms-2384670

Type: Article

Title: ‘Phototherapy of Alzheimer’s Disease During Sleep’ by Semyachkina-Glushkovskaya O et al., submitted to International Journal of Molecular Sciences (IJMS) 

Dear Authors,

One of the current challenges in the field of phototherapy for Alzheimer's disease (AD) is to develop effective and safe methods for delivering light therapy during sleep. Semyachkina-Glushkovskaya and colleagues in the present review article entitled ‘Phototherapy of Alzheimer’s Disease During Sleep’, investigated how photobiomodulation (PBM) of lymphatic drainage and removal system of the brain (LDRSB) during sleep could be used as a new innovative treatment tool in the therapy of AD and other brain pathologies.

The main strength of this paper is that it addresses an interesting and timely question, providing an overview of the current knowledge of the benefits of PBM for AD as a non-invasive, non-pharmaceutical, and safe therapeutic approach. In general, I think the idea of this review is really interesting and the authors’ fascinating observations on this timely topic may be of interest to the readers of IJMS. However, some comments, as well as some crucial evidence that should be included to support the author’s argumentation, needed to be addressed to improve the quality of the manuscript, its adequacy, and its readability prior to the publication in the present form.

Please consider the following comments:

1.      First of all, I would like the authors to address to the following questions and clarify in the manuscript: a) What are the potential benefits of using phototherapy to treat AD during sleep? b) How does phototherapy work to improve cognitive function in Alzheimer's patients? c) Are there any potential risks or side effects associated with phototherapy for AD?

2.      Please consider rewriting the title: in my opinion, in this form, it seems to be not enough informative about the aim of this review. So, I recommend conveying the most important message of this review in the tile.

3.      A graphical abstract that will visually summarize the main findings of the manuscript is highly recommended.

4.      Abstract: According to the Journal’s guidelines, this section should be presented as a short summary of about 200 words maximum that objectively represents the article. It should let readers get the gist or essence of the manuscript quickly, prepare the readers to follow the detailed information, analyses, and arguments in the full paper and, most of all, it should help readers remember key points from your paper. Please, consider rewriting this paragraph following these instructions and proportionally presenting the background, short summary, and conclusion according to the guidelines of the journal (https://www.mdpi.com/journal/ijms/instructions). The background should include the general background (one to two sentences), the specific background (two to three sentences), and current issue addressed to this study (one sentence), leading to the objectives. In this subsection I would like the authors to lay out basic information, problem statement, and the authors’ motivation to break off. The conclusion should include one sentence describing the main result using such words like “Here we highlight”. The conclusion should write the potential and the advance this study has provided in the field and finally a broader perspective (two to three sentences) readily comprehensible to a scientist in any discipline.

5.      I would ask the Authors to clarify the criteria they decided to use for studies’ collection in their review: they should specify the requirements used to decide whether a study met the inclusion/exclusion criteria of the review, describe whether they included a balanced coverage of all information that is actually available, whether they have included the most recent and relevant studies and enough material to show the development and limitations in this field of interest. That said, it is crucial to declare the type of review: systematic, scoping, synthetic, or narrative. Please refer to the checklist for review article the authors intend to and include all elements necessary for a review type (http://prisma-statement.org/PRISMAstatement/checklist.aspx?AspxAutoDetectCookieSupport=1).

6.      The objectives of this study are generally clear and to the point; however, I believe that there are some ambiguous points that require clarification or refining. In my opinion, authors should be explicit regarding how they sought to assess the effects of new photobiomodulation in the treatment of individuals with AD, since this is the key aim of this review.

7.      PBM in clinical studies: In this section, authors focused on describing mechanisms of PBM effects in clinical studies. In this regard, I would ask the authors to also provide more information about studies that explored the potential benefits of PBM on lymphatic pumping and contractility, which are considered the main physiological mechanisms underlying fluid transport and waste clearance from tissues thereby providing evidence that transcranial and remote PBM can augment the drainage and clearance function of lymphatic vessels, providing a therapeutic target for neurodegenerative diseases (DOI: 10.3390/biomedicines10122999; DOI: 10.3390/cells11162607).

8.      Although not mandatory, I believe that a final ‘Discussion’ section could be very useful to capture the state of art well, and in this respect, I would like to see in this section some views on a way forward, for example some good discussion on human and animal studies that have investigated the application of new non-invasive and non-pharmaceutical therapies in AD treatment, for example by using Non-invasive brain stimulation techniques (NIBS). Indeed, Authors could discuss knowledge of NIBS-induced network effects that could be used to optimize NIBS therapy in age-related neurodegenerative diseases characterized by progressive neural network disruption (DOI: 10.3390/ijms24065926). So, I recommend that the authors reorganize this section with up to 1500 words, clarifying the following essential elements for discussion. Starting with an introductory paragraph, I would like the authors to present the summary of the previous section and to develop argument on the potential of this study complementing as the extension of the previous work, the implication of the findings of this review, how this study could facilitate future research, the ultimate goal, the challenge, the knowledge and the technology necessary to achieve this goal, the statement about this field in general, and finally the importance of this line of research.

9.      I would ask the authors to include a proper and defined ‘Limitations and future directions’ section before the end of the manuscript, in which authors can describe in detail and report all the technical issues that could be brought to the surface.

10.  Conclusion: I believe that presenting the conclusion section would benefit from a single paragraph presenting some thoughtful as well as in-depth considerations by the authors as experts to convey the take-home message, as it is very descriptive but not enough theoretical as a conclusion should be. The authors should make their effort to explain the theoretical implication as well as the translational application of their research.

11.  Tables: According to the Journal’s guidelines, please provide a short explanatory caption for the table within the text.

12.  References: Authors should consider revising the bibliography, as there are several incorrect citations. Indeed, according to the Journal’s guidelines, they should provide the abbreviated journal name in italics, the year of publication in bold, the volume number in italics for all the references. Please cite more references. Review articles like this typically cite more than 150 references.

Overall, the manuscript contains two figures, two tables, and 133 references. I believe that the manuscript may carry important value in providing valuable insights into the potential of phototherapy during sleep as a promising approach for the treatment of AD, which could lead to significant improvements in patient outcomes. I hope that, after these careful revisions, the manuscript can meet the Journal’s high standards for publication. I am available for a new round of revision of this article.

Best regards,

Reviewer 

The quality of English is average.

Author Response

Comments: First of all, I would like the authors to address to the following questions and clarify in the manuscript: a) What are the potential benefits of using phototherapy to treat AD during sleep? b) How does phototherapy work to improve cognitive function in Alzheimer's patients? c) Are there any potential risks or side effects associated with phototherapy for AD?

Response: The authors would like to express their sincere gratitude for the positive assessment of our review and for the important recommendations for improving its quality. We added information about the benefits of using of PBM for AD during sleep, relevant standards of safe use of PBM and its possible side-effects as well as PBM-mediated improvement of cognitive funtions in the subjects with AD in Discussion (Lines 322-427). All changes in the text of the article are highlighted in yellow.

Comment: Please consider rewriting the title: in my opinion, in this form, it seems to be not enough informative about the aim of this review. So, I recommend conveying the most important message of this review in the tile.

Response: We improved the title of manuscript (Lines 1-3).

Comment: A graphical abstract that will visually summarize the main findings of the manuscript is highly recommended.

Response: We added a graphical abstract, which is at the end of the manuscript.

Comment: Abstract: According to the Journal’s guidelines, this section should be presented as a short summary of about 200 words maximum that objectively represents the article. It should let readers get the gist or essence of the manuscript quickly, prepare the readers to follow the detailed information, analyses, and arguments in the full paper and, most of all, it should help readers remember key points from your paper. Please, consider rewriting this paragraph following these instructions and proportionally presenting the background, short summary, and conclusion according to the guidelines of the journal (https://www.mdpi.com/journal/ijms/instructions). The background should include the general background (one to two sentences), the specific background (two to three sentences), and current issue addressed to this study (one sentence), leading to the objectives. In this subsection I would like the authors to lay out basic information, problem statement, and the authors’ motivation to break off. The conclusion should include one sentence describing the main result using such words like “Here we highlight”. The conclusion should write the potential and the advance this study has provided in the field and finally a broader perspective (two to three sentences) readily comprehensible to a scientist in any discipline.

Response: We improved Abstract (Lines 16-29).

Comment: I would ask the Authors to clarify the criteria they decided to use for studies’ collection in their review: they should specify the requirements used to decide whether a study met the inclusion/exclusion criteria of the review, describe whether they included a balanced coverage of all information that is actually available, whether they have included the most recent and relevant studies and enough material to show the development and limitations in this field of interest. That said, it is crucial to declare the type of review: systematic, scoping, synthetic, or narrative. Please refer to the checklist for review article the authors intend to and include all elements necessary for a review type (http://prisma-statement.org/PRISMAstatement/checklist.aspx?AspxAutoDetectCookieSupport=1).

Response: We added the description of search strategy for our review (Lines 57-91).

Comments: The objectives of this study are generally clear and to the point; however, I believe that there are some ambiguous points that require clarification or refining. In my opinion, authors should be explicit regarding how they sought to assess the effects of new photobiomodulation in the treatment of individuals with AD, since this is the key aim of this review.

PBM in clinical studies: In this section, authors focused on describing mechanisms of PBM effects in clinical studies. In this regard, I would ask the authors to also provide more information about studies that explored the potential benefits of PBM on lymphatic pumping and contractility, which are considered the main physiological mechanisms underlying fluid transport and waste clearance from tissues thereby providing evidence that transcranial and remote PBM can augment the drainage and clearance function of lymphatic vessels, providing a therapeutic target for neurodegenerative diseases (DOI: 10.3390/biomedicines10122999; DOI: 10.3390/cells11162607).

Although not mandatory, I believe that a final ‘Discussion’ section could be very useful to capture the state of art well, and in this respect, I would like to see in this section some views on a way forward, for example some good discussion on human and animal studies that have investigated the application of new non-invasive and non-pharmaceutical therapies in AD treatment, for example by using Non-invasive brain stimulation techniques (NIBS). Indeed, Authors could discuss knowledge of NIBS-induced network effects that could be used to optimize NIBS therapy in age-related neurodegenerative diseases characterized by progressive neural network disruption (DOI: 10.3390/ijms24065926). So, I recommend that the authors reorganize this section with up to 1500 words, clarifying the following essential elements for discussion. Starting with an introductory paragraph, I would like the authors to present the summary of the previous section and to develop argument on the potential of this study complementing as the extension of the previous work, the implication of the findings of this review, how this study could facilitate future research, the ultimate goal, the challenge, the knowledge and the technology necessary to achieve this goal, the statement about this field in general, and finally the importance of this line of research.

I would ask the authors to include a proper and defined ‘Limitations and future directions’ section before the end of the manuscript, in which authors can describe in detail and report all the technical issues that could be brought to the surface.

Response: Many thanks for the constructive advices that we took into account in Discussion (Lines 322-427).

Comment: Conclusion: I believe that presenting the conclusion section would benefit from a single paragraph presenting some thoughtful as well as in-depth considerations by the authors as experts to convey the take-home message, as it is very descriptive but not enough theoretical as a conclusion should be. The authors should make their effort to explain the theoretical implication as well as the translational application of their research.

Response: We improved Conclusion (Lines 430-463).

Comment: Tables: According to the Journal’s guidelines, please provide a short explanatory caption for the table within the text.

Response: We added a short description of tables 1 and 2 (Lines 228-230; 248-250)

Comment: References: Authors should consider revising the bibliography, as there are several incorrect citations. Indeed, according to the Journal’s guidelines, they should provide the abbreviated journal name in italics, the year of publication in bold, the volume number in italics for all the references. Please cite more references. Review articles like this typically cite more than 150 references.

Response: We corrected References. The total number of citations is 170 in the improved manuscript.

The authors thank the referee for an interest in our review and for the opportunity to improve its quality with the helpful advices.

Authors

Round 2

Reviewer 2 Report

12 May 2023 

Manuscript ID: ijms-2384670

Type: Article

Title: ‘Phototherapy of Alzheimer’s Disease During Sleep’ by Semyachkina-Glushkovskaya O et al., submitted to International Journal of Molecular Sciences (IJMS) 

Dear Authors,

I am pleased to see that the authors have addressed many of the concerns I raised during the previous round of peer review. However, the revisions are incomplete; therefore, the authors must revise the manuscript to meet the publication requirements of the journal. I would like the authors to pay close attention to the following comments and parts I have highlighted: "..." Also, I would like the authors to present a pointwise rebuttal clarifying the authors' revision.

Please consider the following comments:

1.      First, I would like the authors to present all essential components of a systematic review, including the PRISMA flowchart and risk of bias assessment. Please visit http://prisma-statement.org/PRISMAstatement/checklist.aspx?AspxAutoDetectCookieSupport=1 for additional information. In addition, please be familiar with the article structure of IJMS (Introduction, Results, Discussion, Methods, and Conclusion).

2.      Please use a short, self-explanatory title that captures the most important message of this review. Suggestion: "Illuminating the Path to Alzheimer's Disease Treatment: Photobiomodulation of Brain Lymphatics during Sleep". Reference: https://plos.org/resource/how-to-write-a-great-title/.

3.      Abstract: I recommend the authors reorganize this section with 200–220 words, proportionally presenting the following subsections without headings: the background, a short summary, and the conclusion. The background should include the general background (one to two sentences), the specific background (two to three sentences), and “the current issue addressed by this review (one sentence)”, leading to “the objectives.” In this subsection, I would like the authors to lay out basic information, a problem statement, and their motivation to break off. The short summary ends with a sentence that puts this subsection in a general context. The conclusion should include one sentence describing the main result using words like “Here we highlight”. “I would like the authors to highlight main findings of this systematic review.” The conclusion should describe the potential and the advance this study has provided in the field, and finally, a broader perspective (two to three sentences) readily comprehensible to a scientist in any discipline.

4.      Keywords: Please list “ten” keywords chosen from “Medical Subject Headings (MeSH)” (https://meshb.nlm.nih.gov/) and use as many as possible in the title and in the first two sentences of the abstract.

5.      Introduction: The authors need to fully develop this section with about “1000 words” and several paragraphs, introducing information on “the key study constructs” that should be understood by readers in any discipline, and make it persuasive enough to advance the primary goal of the author's recent research and the particular goal the author has intended by this review. I'd like to suggest that the authors present the introduction beginning with the overall context, moving on to the specific context, and concluding with the current problem addressed in this study before moving on to the objectives. Those key structures ought to be set up logically and coherently.

6.      PBM in clinical studies: In this section, authors focused on describing mechanisms of PBM effects in clinical studies. In this regard, I would ask the authors to also provide more information, providing a therapeutic target for neurodegenerative diseases (DOI: 10.3390/biomedicines10122999; DOI: 10.3390/cells11162607).

7.      Discussion: I recommend that the authors reorganize this section with up to 1500 words, clarifying the following essential elements for discussion. Starting with an introductory paragraph, I would like the authors to present the summary of the previous section and to develop argument on the potential of this study complementing as the extension of the previous work, the implication of the findings of this review, how this study could facilitate future research, the ultimate goal, the challenge, the knowledge and the technology necessary to achieve this goal, the statement about this field in general, and finally the importance of this line of research. In this regard, I would like to see in this section some views on a way forward, for example some good discussion on human and animal studies that have investigated the application of new non-invasive and non-pharmaceutical therapies in Alzheimer’s disease treatment, for example by using Non-invasive brain stimulation techniques (NIBS). Indeed, Authors could discuss knowledge of NIBS-induced network effects that could be used to optimize NIBS therapy in age-related neurodegenerative diseases characterized by progressive neural network disruption (DOI: 10.3390/ijms24065926).

8.      Conclusion: I believe that presenting the conclusion section would benefit from “a single paragraph” presenting some thoughtful as well as in-depth considerations by the authors as experts to convey the take-home message, as it is very descriptive but not enough theoretical as a conclusion should be. The authors should make their effort to explain the theoretical implication as well as the translational application of their research.

9.      References: Please follow the guidelines of the journal (https://www.mdpi.com/journal/ijms/instructions) and provide it with doi number.  

Overall, the manuscript contains two figures, two tables, and 170 references. I believe that the manuscript may carry important value in providing valuable insights into the potential of phototherapy during sleep as a promising approach for the treatment of AD, which could lead to significant improvements in patient outcomes. I hope that, after these careful revisions, the manuscript can meet the Journal’s high standards for publication. I am available for a new round of revision of this article.

Best regards,

Reviewer

Manuscript ID: ijms-2384670

Type: Article

Title: ‘Phototherapy of Alzheimer’s Disease During Sleep’ by Semyachkina-Glushkovskaya O et al., submitted to International Journal of Molecular Sciences (IJMS) 

Dear Ms. Nicole Xiong, 

After reviewing the document, it is evident that minor editing of the English language is required. The document contains several grammatical errors. Additionally, some sentences are unclear and require rephrasing to improve readability. While the document's content is informative and well organized, the English language's quality needs improvement to ensure that the document is clear and concise. Therefore, minor editing is necessary to improve the document's overall quality and readability. 

Best regards,  

Reviewer

Author Response

Comments: First, I would like the authors to present all essential components of a systematic review, including the PRISMA flowchart and risk of bias assessment. Please visit http://prisma-statement.org/PRISMAstatement/checklist.aspx?AspxAutoDetectCookieSupport=1 for additional information. In addition, please be familiar with the article structure of IJMS (Introduction, Results, Discussion, Methods, and Conclusion).

Response: The authors are very grateful to the referee for constructive advices in improving the structure and style of our review. We added the flowchart (Figure 1) and all headings of the article, including Introduction, Methods, Results, Discussion, and Conclusion. Our review is aimed at highlighting a completely new direction in the treatment of Alzheimer's disease, which is based on the use of phototherapy. There are very few completed clinical studies with reliable results in this area. We show all of them in Tables 1 and 2. We also discuss pioneering studies showing that sleep phototherapy is more effective in stimulating the elimination of toxins from the brain, including beta amyloid. This direction is also in its infancy, which began to be actively cited after our first works. Considering the pioneering directions covered in our review  and the limited literature in this field, our review does not quite meet the criteria for a systematic review. Our state of the art review provides a time-based overview of the current state of knowledge about the photherapy of Alzheimer's disease during sleep and suggests directions for future research. We may partially use the elements of PRISMA as a flowchart that visually reflects the literature search strategy, inclusion and exclusion criteria, but we do not conduct a meta-analysis and do not give a risk of bias assessment.

All corrections in the text are highlighted in yellow.

Comment: Please use a short, self-explanatory title that captures the most important message of this review. Suggestion: "Illuminating the Path to Alzheimer's Disease Treatment: Photobiomodulation of Brain Lymphatics during Sleep". Reference: https://plos.org/resource/how-to-write-a-great-title/.

Response: We improved the title of review (Lines 2 and 3).

"Phototherapy of Alzheimer's Disease: Photostimulation of Brain Lymphatics during Sleep".

The title of the article reflects the two main directions of our review: 1) the phototherapy of Alzheimer's disease; 2) the phototherapy of Alzheimer's disease during deep sleep. We explain this by the fact that the brain lymphatic system is activated during deep sleep, and photoinfluence on the lymphatic vessels at the time of their natural activation contributes to a better removal of toxins and beta-amyloid from the brain tissues, which underlies the increase in resistance to the progression of the disease.

Comment: Abstract: I recommend the authors reorganize this section with 200–220 words, proportionally presenting the following subsections without headings: the background, a short summary, and the conclusion. The background should include the general background (one to two sentences), the specific background (two to three sentences), and “the current issue addressed by this review (one sentence)”, leading to “the objectives.” In this subsection, I would like the authors to lay out basic information, a problem statement, and their motivation to break off. The short summary ends with a sentence that puts this subsection in a general context. The conclusion should include one sentence describing the main result using words like “Here we highlight”. “I would like the authors to highlight main findings of this systematic review.” The conclusion should describe the potential and the advance this study has provided in the field, and finally, a broader perspective (two to three sentences) readily comprehensible to a scientist in any discipline.

Response: We improved the abstract (Lines 16-29).

Comment: Keywords: Please list “ten” keywords chosen from “Medical Subject Headings (MeSH)” (https://meshb.nlm.nih.gov/) and use as many as possible in the title and in the first two sentences of the abstract.

Response: We added ten keywords that reflects all the topics covered in our review (Lines 30, 31).

Comment: Introduction: The authors need to fully develop this section with about “1000 words” and several paragraphs, introducing information on “the key study constructs” that should be understood by readers in any discipline, and make it persuasive enough to advance the primary goal of the author's recent research and the particular goal the author has intended by this review. I'd like to suggest that the authors present the introduction beginning with the overall context, moving on to the specific context, and concluding with the current problem addressed in this study before moving on to the objectives. Those key structures ought to be set up logically and coherently.

Response: We improved the introduction and added more explanation about the choice of the main idea of our review article highlighting a new direction in the treatment of Alzheimer's disease based on the use of photobiomodulation during sleep (Lines 34-89).

Comment: PBM in clinical studies: In this section, authors focused on describing mechanisms of PBM effects in clinical studies. In this regard, I would ask the authors to also provide more information, providing a therapeutic target for neurodegenerative diseases (DOI: 10.3390/biomedicines10122999; DOI: 10.3390/cells11162607)

Response: Thank you so much for this advice. We added the session "PBM for other brain diseases", where we show that PBM has been shown to be effective in the therapy of other brain diseases, including Parkinson’s disease, amyotrophic lateral sclerosis and multiple sclerosis, autism and hypoxic-ischemic brain injuries caused by cardiac arrest and hypoxic-ischemic encephalopathy (Lines 372-423).

Comment: Discussion: I recommend that the authors reorganize this section with up to 1500 words, clarifying the following essential elements for discussion. Starting with an introductory paragraph, I would like the authors to present the summary of the previous section and to develop argument on the potential of this study complementing as the extension of the previous work, the implication of the findings of this review, how this study could facilitate future research, the ultimate goal, the challenge, the knowledge and the technology necessary to achieve this goal, the statement about this field in general, and finally the importance of this line of research. In this regard, I would like to see in this section some views on a way forward, for example some good discussion on human and animal studies that have investigated the application of new non-invasive and non-pharmaceutical therapies in Alzheimer’s disease treatment, for example by using Non-invasive brain stimulation techniques (NIBS). Indeed, Authors could discuss knowledge of NIBS-induced network effects that could be used to optimize NIBS therapy in age-related neurodegenerative diseases characterized by progressive neural network disruption (DOI: 10.3390/ijms24065926).

Response: We improved the discussion and increased the text to 1544 words (Lines 465-585).

Comment: Conclusion: I believe that presenting the conclusion section would benefit from “a single paragraph” presenting some thoughtful as well as in-depth considerations by the authors as experts to convey the take-home message, as it is very descriptive but not enough theoretical as a conclusion should be. The authors should make their effort to explain the theoretical implication as well as the translational application of their research.

Response: We improved the conclusion and reduced it to one paragraph (Lines 589-603).

Comment: References: Please follow the guidelines of the journal (https://www.mdpi.com/journal/ijms/instructions) and provide it with doi number.  

Response: We used the ijms-template for preparation of our review. In accordance with the recommendations, doi numbers are not provided in the references:

  1. Author 1, A.B.; Author 2, C.D. Title of the article. Abbreviated Journal Name Year, Volume, page range.
  2. Author 1, A.; Author 2, B. Title of the chapter. In Book Title, 2nd ed.; Editor 1, A., Editor 2, B., Eds.; Publisher: Publisher Location, Country, 2007; Volume 3, pp. 154–196.
  3. Author 1, A.; Author 2, B. Book Title, 3rd ed.; Publisher: Publisher Location, Country, 2008; pp. 154–196.

The authors express their sincere gratitude to the referee for the interest in our review and for the useful advices that helped to significantly improve our article for its possible publication in International Journal of Molecular Sciences.

Round 3

Reviewer 2 Report

2 June 2023 

Manuscript ID: ijms-2384670

Type: Article

Title: ‘Phototherapy of Alzheimer’s Disease During Sleep’ by Semyachkina-Glushkovskaya O et al., submitted to International Journal of Molecular Sciences (IJMS) 

Dear Authors,

I acknowledge that the authors have made revisions to the manuscript, but the majority of the issues I brought up during the previous round of peer review have not been adequately addressed. In order to meet the high standards for publication set by the journal, I would like the authors to pay close attention to the inclusion of the essential components of review articles, particularly as a systematic review, and to my comments below. Again, I sincerely hope I can assist the authors in having them focus particularly on the remarks and passages I have underlined: "..." Also, I would like the authors to present a pointwise rebuttal fully clarifying “the authors' revision” in addition to their comments.

Comments:

1.      First, I would like the authors to present all essential components of a systematic review, including the PRISMA flowchart and “risk of bias assessment” [1]. The authors ought to declare “systematic review” in the abstract and the objectives of the introduction.  In addition, please be familiar with “the article structure of IJMS” (Introduction, “Results”, Discussion, “Methods”, and Conclusion).

2.      Abstract: Please pay close attention to my previous comments: I recommend the authors reorganize this section with 200–220 words, proportionally presenting the following subsections without headings: the background, a short summary, and the conclusion. The background should include the general background (one to two sentences), the specific background (two to three sentences), and “the current issue addressed by this review (one sentence)”, leading to “the objectives.” In this subsection, I would like the authors to lay out basic information, a problem statement, and their motivation to break off. The short summary ends with a sentence that puts this subsection in a general context. The conclusion should include one sentence describing the main result using words like “Here we highlight”. “I would like the authors to highlight main findings of this systematic review.” The conclusion should describe the potential and the advance this study has provided in the field, and finally, a broader perspective (two to three sentences) readily comprehensible to a scientist in any discipline [2-4].

3.      Keywords: A majority of the keywords listed by the authors are not listed in the Medical Subject Headings (MeSH). Please list “ten” keywords “chosen” from “MeSH” (https://meshb.nlm.nih.gov/) and use as many as possible in the title and in the first two sentences of the abstract. This is the crucial points to get attention of readers and thus a chance of being cited [5].

4.      Introduction: Please try to focus on the structure of the introduction according to my following comments: The authors need to fully develop this section with about “1000 words” and several paragraphs, introducing information on “the key study constructs” that should be understood by readers in any discipline, and make it persuasive enough to advance the primary goal of the author's recent research and the particular goal the author has intended by this review. I'd like to suggest that the authors present the introduction beginning with the overall context, moving on to the specific context, and concluding with the current problem addressed in this study before moving on to the objectives. Those key structures ought to be set up logically and coherently [6].

5.      In this regard, I would like the authors to focused on describing mechanisms of PBM effects in clinical studies and to also provide more information, providing a therapeutic target for neurodegenerative diseases.  The following works may enhance the value of this manuscript, including but not limited to: DOI: 10.3390/biomedicines10122999; DOI: 10.3390/cells11162607.

6.      Results: I suggest “closing this section with a paragraph which puts the results into a more general context.”

7.      Discussion: Please try to focus on the structure of the introduction according to my following comments: I recommend that the authors reorganize this section with up to 1500 words, clarifying the following essential elements for discussion. Starting with an introductory paragraph, I would like the authors to present the summary of the previous section and to develop argument on the potential of this study complementing as the extension of the previous work, the implication of the findings of this review, how this study could facilitate future research, the ultimate goal, the challenge, the knowledge and the technology necessary to achieve this goal, the statement about this field in general, and finally the importance of this line of research. In this regard, I would like to see in this section some views on a way forward, for example some good discussion on human and animal studies that have investigated the application of new non-invasive and non-pharmaceutical therapies in Alzheimer’s disease treatment, for example by using Non-invasive brain stimulation techniques (NIBS). Indeed, Authors could discuss knowledge of NIBS-induced network effects that could be used to optimize NIBS therapy in age-related neurodegenerative diseases characterized by progressive neural network disruption (DOI: 10.3390/ijms24065926) [7,8].

8.      Conclusion: Please declare “systematic”: I believe that presenting the conclusion section would benefit from “a single paragraph” presenting some thoughtful as well as in-depth considerations by the authors as experts to convey the take-home message, as it is very descriptive but not enough theoretical as a conclusion should be. The authors should make their effort to explain the theoretical implication as well as the translational application of their research.

9.      References: Please follow the guidelines of the journal (https://www.mdpi.com/journal/ijms/instructions):  journal abbreviation is punctuated with a period and please provide it with “doi number”.  

Overall, the manuscript contains two figures, two tables, and 179 references. I believe that the manuscript may carry important value in providing valuable insights into the potential of phototherapy during sleep as a promising approach for the treatment of Alzheimer’s disease, which could lead to significant improvements in patient outcomes. I hope that, after these careful revisions, the manuscript can meet the Journal’s high standards for publication. I am available for a new round of revision of this article.

Best regards,

Reviewer

References:

1.      https://www.equator-network.org/reporting-guidelines/prisma/

2.      https://www.scribbr.com/dissertation/abstract/

3.      https://writing.wisc.edu/handbook/assignments/writing-an-abstract-for-your-research-paper/

4.      https://pubmed.ncbi.nlm.nih.gov/30930712/

5.      https://www.ncbi.nlm.nih.gov/pmc/articles/PMC7144240/

6.      https://dept.writing.wisc.edu/wac/writing-an-introduction-for-a-scientific-paper/

7.      https://www.ncbi.nlm.nih.gov/pmc/articles/PMC4404856/

8.      https://www.scribbr.com/dissertation/discussion/

2 June 2023 

Manuscript ID: ijms-2384670

Type: Article

Title: ‘Phototherapy of Alzheimer’s Disease During Sleep’ by Semyachkina-Glushkovskaya O et al., submitted to International Journal of Molecular Sciences (IJMS) 

Dear Ms. Nicole Xiong, 

After reviewing the document, it is evident that minor editing of the English language is required. The document contains several grammatical errors, including incorrect verb tense, subject-verb agreement, and punctuation errors. Additionally, some sentences are unclear and require rephrasing to improve readability. While the document's content is informative and well organized, the English language's quality needs improvement to ensure that the document is clear and concise. Therefore, minor editing is necessary to improve the document's overall quality and readability.

Best regards,

Reviewer

Author Response

Comments: First, I would like the authors to present all essential components of a systematic review, including the PRISMA flowchart and “risk of bias assessment” [1]. The authors ought to declare “systematic review” in the abstract and the objectives of the introduction.  In addition, please be familiar with “the article structure of IJMS” (Introduction, “Results”, Discussion, “Methods”, and Conclusion).

ReferencesPlease follow the guidelines of the journal (https://www.mdpi.com/journal/ijms/instructions):  journal abbreviation is punctuated with a period and please provide it with “doi number”.  

Response: The authors thank the referee for the opportunity to improve our paper and useful comments. Our review is devoted to a pioneering direction, in which there are not yet a sufficient number of publications to classify our review as a systematic one. We strongly believe that the structure of a review article is determined by the strategy of its idea. We published several our review articles in the Int J Mol Sci, including the invited reviews and the highly cited reviews in MDPI (Int. J. Mol. Sci. 2023, 24, 3221. https://doi.org/10.3390/ijms24043221; Int. J. Mol. Sci. 2021, 22, 6917. https://doi.org/10.3390/ijms22136917; Int. J. Mol. Sci. 2020, 21, 6293. https://doi.org/10.3390/ijms21176293; Int. J. Mol. Sci. 2018, 19, 3818. https://doi.org/10.3390/ijms19123818). We strongly hope that our review in its current form meets the requirements of the journal. We used the MDPI template that is recommended for the design of articles. As we have already noted, there is no need to cite the doi articles.

The innovative direction covered in our review and the limited number of publications in this new direction, as well as the type "State of the art review", make the statistical analysis or risk of bias assessment” not objective, which is far from the main goal of our review.

Comment: Abstract: Please pay close attention to my previous comments: I recommend the authors reorganize this section with 200–220 words, proportionally presenting the following subsections without headings: the background, a short summary, and the conclusion. The background should include the general background (one to two sentences), the specific background (two to three sentences), and “the current issue addressed by this review (one sentence)”, leading to “the objectives.” In this subsection, I would like the authors to lay out basic information, a problem statement, and their motivation to break off. The short summary ends with a sentence that puts this subsection in a general context. The conclusion should include one sentence describing the main result using words like “Here we highlight”. “I would like the authors to highlight main findings of this systematic review.” The conclusion should describe the potential and the advance this study has provided in the field, and finally, a broader perspective (two to three sentences) readily comprehensible to a scientist in any discipline [2-4].

Response: Abstracts include the necessary information, such as the the background, a short summary, and the conclusion as well as Abstract includes 197 words.

The background: The global number of people with Alzheimer’s disease (AD) doubles every 5 years. It has been established that unless an effective treatment for AD is found, the incidence of AD will triple by 2060. However, pharmacological therapies of AD have failed to show effectiveness and safety. Therefore, the search for alternative methods of AD treating is an urgent problem of medicine. 

The main idea: The lymphatic drainage and removal system of the brain (LDRSB) plays an important role in resistance to the progression of AD. The development of methods for augmentation of the LDRSB functions may contribute to progress in the AD therapy. The photobiomodulation (PBM) is considered as a non-pharmacological and safe approach for the AD therapy. Here, we highlight the most recent and relevant studies of PBM for AD. We focus on emerging evidence that indicates the potential benefits of PBM during sleep for modulation of natural activation of the LDRSB during night providing effective removal of metabolites, including amyloid-β, from the brain leading to reduced progression of AD.  

The conclusion: Our review open a new niche in the therapy of brain diseases during sleep and shed light on the perspectives of development of smart sleep technologies for neurodegenerative diseases. 

Comment: Keywords: A majority of the keywords listed by the authors are not listed in the Medical Subject Headings (MeSH). Please list “ten” keywords “chosen” from “MeSH” (https://meshb.nlm.nih.gov/) and use as many as possible in the title and in the first two sentences of the abstract. This is the crucial points to get attention of readers and thus a chance of being cited [5].

Response: We have included those words and phrases that best suit the direction of the review, its title and the main idea. We hope that for this it is not necessary to take words from the Medical Subject Headings, in which there are no necessary words close to our review.

Comment: Introduction: Please try to focus on the structure of the introduction according to my following comments: The authors need to fully develop this section with about “1000 words” and several paragraphs, introducing information on “the key study constructs” that should be understood by readers in any discipline, and make it persuasive enough to advance the primary goal of the author's recent research and the particular goal the author has intended by this review. I'd like to suggest that the authors present the introduction beginning with the overall context, moving on to the specific context, and concluding with the current problem addressed in this study before moving on to the objectives. Those key structures ought to be set up logically and coherently [6].

Response: The introduction includes 743 words, in which the direction of the main idea of the review and its key points have been cut off. There are no strict recommendations in the rules for writing review articles that the introduction should include 1000 words. We hope that the number of words in any section of the review is determined by logic, but not by simple following to get the required number of words. An artificial increase in the volume of the introduction can lead to the fact that readers lose the main idea and interest in the review.

Comment: In this regard, I would like the authors to focused on describing mechanisms of PBM effects in clinical studies and to also provide more information, providing a therapeutic target for neurodegenerative diseases.  The following works may enhance the value of this manuscript, including but not limited to: DOI: 10.3390/biomedicines10122999; DOI: 10.3390/cells11162607.

Response: We included already the section "PBM for other brain diseases" in our previous improved version of manuscript (Lines 372-423). The idea of our review is to show the therapeutic effects of PBM for Alzheimer's disease through photostimulation of the lymphatic system of the brain during sleep. The articles you recommend (DOI: 10.3390/biomedicines10122999; DOI: 10.3390/cells11162607) are far from the main idea of our review. They are not related to phototherapy or the lymphatics or sleep. Can we ask you why you recommend them?

Comments: 

Results: I suggest “closing this section with a paragraph which puts the results into a more general context.”

DiscussionPlease try to focus on the structure of the introduction according to my following comments: I recommend that the authors reorganize this section with up to 1500 words, clarifying the following essential elements for discussion. Starting with an introductory paragraph, I would like the authors to present the summary of the previous section and to develop argument on the potential of this study complementing as the extension of the previous work, the implication of the findings of this review, how this study could facilitate future research, the ultimate goal, the challenge, the knowledge and the technology necessary to achieve this goal, the statement about this field in general, and finally the importance of this line of research. In this regard, I would like to see in this section some views on a way forward, for example some good discussion on human and animal studies that have investigated the application of new non-invasive and non-pharmaceutical therapies in Alzheimer’s disease treatment, for example by using Non-invasive brain stimulation techniques (NIBS). Indeed, Authors could discuss knowledge of NIBS-induced network effects that could be used to optimize NIBS therapy in age-related neurodegenerative diseases characterized by progressive neural network disruption (DOI: 10.3390/ijms24065926) [7,8].

Response: The results and discussion are interrelated structures of the review article. Therefore, we begin the discussion with an introductory paragraph, continuing to discuss the mechanisms of PBM for Alzheimer's disease, how it can increase resistance to disease development and affect memory. We discuss the role of the brain's lymphatic system in Alzheimer's phototherapy as an important mechanism for the removal of toxins from the brain, including beta-amyloid. We also discuss the technologies required to improve the quality of phototherapy. To do this, we discuss sleep as a natural state of activation of the brain's lymphatic drainage function and show what technologies need to be created (combining PBM and sleep detectors) in order to effectively treat Alzheimer's disease. Such a logic will allow readers in the Results to know about the pioneering works in the direction of our review, and in the Discussion we give the new knowledge that will help to understand better how non-invasive phototherapy can have therapeutic effects on Alzheimer's disease through the activation of the brain's lymphatic system.

Comment: Conclusion: Please declare “systematic”: I believe that presenting the conclusion section would benefit from “a single paragraph” presenting some thoughtful as well as in-depth considerations by the authors as experts to convey the take-home message, as it is very descriptive but not enough theoretical as a conclusion should be. The authors should make their effort to explain the theoretical implication as well as the translational application of their research.

Response: Our review is not a systematic review. Type of our review - State of the art review. The logic of the conclusion is to guide the reader along the main focus of the review, namely phototherapy, lymphatics, sleep, and technologies that combine PBM and sleep detection. It is this unconventional approach that can be an exciting impulse for readers, opening new doors to the development of breakthrough technologies for the treatment of Alzheimer's disease. In this sense, the type of our review is close to the rules of art - do not follow trivial paths and follow creative paths to spark interest and imagination.

Many thanks for your interest in our review and our experience in the study of structure of review articles. This is a great experience for us and we are sincerely grateful to you for this

Authors.
